# Hybrid 8-bit Floating Point (HFP8) Training and Inference for Deep Neural Networks

Xiao Sun        Jungwook Choi*        Chia-Yu Chen        Naigang Wang

Swagath Venkataramani        Vijayalakshmi Srinivasan        Xiaodong Cui        Wei Zhang

**Kailash Gopalakrishnan**
IBM T. J. Watson Research Center
Yorktown Heights, NY 10598, USA
{xsun, * ,cchen,nwang,swagath.venkataramani,viji,cuix,weiz,kailash}@us.ibm.com

## Abstract

Reducing the numerical precision of data and computation is extremely effective in accelerating deep learning training workloads. Towards this end, 8-bit floating point representations (FP8) were recently proposed for DNN training. However, its applicability was only demonstrated on a few selected models and significant degradation is observed when popular networks such as MobileNet and Transformer are trained using FP8. This degradation is due to the inherent precision requirement difference in the forward and backward passes of DNN training. Using theoretical insights, we propose a hybrid FP8 (HFP8) format and DNN end-to-end distributed training procedure. We demonstrate, using HFP8, the successful training of deep learning models across a whole spectrum of applications including Image Classification, Object Detection, Language and Speech without accuracy degradation. Finally, we demonstrate that, by using the new 8 bit format, we can directly quantize a pre-trained model down to 8-bits without losing accuracy by simply fine-tuning batch normalization statistics. These novel techniques enable a new generations of 8-bit hardware that are robust for building and deploying neural network models.

## 1   Introduction

As Deep Neural Networks (DNNs) evolve rapidly and as models get more complex, training times have increased significantly. To mitigate this challenge, efficient training through reduced precision exploitation has become increasingly important. Using reduced precision for data representations and general matrix multiplications (GEMM) can accelerate DNN training dramatically and save significant computing time and power. Indeed, GPUs can already perform mixed-precision training with 16-bit IEEE Half-Precision floating point formats for deep learning tasks [1]. Recently, a new (1-5-2) (sign-exponent-mantissa) floating-point 8-bit format (FP8), was used to successfully train popular ImageNet models [2] without much accuracy loss. In addition, 8-bit Fixed point formats (INT8) have also been explored to train ResNet50 successfully although 1 of the 3 GEMM computations was performed in higher precision [3]. In addition to DNN training, efficient low-precision deployment is critical in a wide range of edge inference use cases where cost and energy constraints can limit

performance [4]. Towards that end, Trans-Precision inference, where models are trained in higher precision and deployed in lower precision formats, have become extremely important [5, 6, 7].

While 8-bit training techniques have progressed rapidly, recent work [2, 3, 8, 9] have only demonstrated its applicability on a small subset of deep learning models—focused around convolution networks such as ResNet [10]. Indeed, plethora of challenges exist to extend FP8 training to a broader spectrum of applications such as image classification, object detection, speech and natural language processing while preserving model accuracy. Furthermore, in large-scale distributed training systems, FP8 acceleration of GEMM and Convolution operations within each learner makes the communication between learners at the weight update step a critical bottleneck. Alleviating this bottleneck using 8-bit communication schemes could substantially improve the end-to-end training performance for distributed DNN training. In addition, for low-precision inference, fixed point techniques involving costly retraining of networks for ultra-short bit-widths [11, 12, 13] as well as post-training quantization for simpler deployment of the INT8/INT4 inference models [7, 14, 15, 16] have been extensively explored, but the state-of-the-art techniques still lose significant model accuracy when they are applied to compact models like MobileNet [17] on large datasets (e.g., ImageNet). In comparison to the fixed-point representation, FP8 based schemes have a wider dynamic range and do not need to find the right quantization range for each layer and channel—serving post-training quantization more naturally.

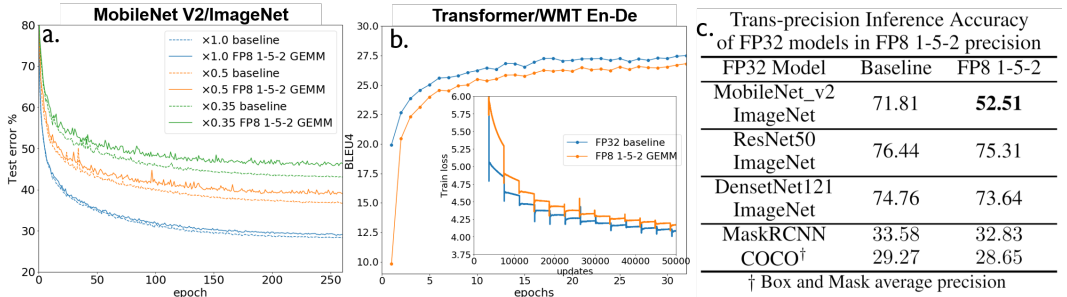

Figure 1: Challenges in the previous FP8 (1-5-2) format. (a) FP8 training on MobileNetV2 with different width multipliers—showcasing significant accuracy degradation from FP8 training for capacity constrained models. (b) FP8 training on Transformer-based machine translation — large loss in BLEU scores. (c) Table for Trans-Precision inference from FP32 to FP8 — large accuracy loss observed in various domains.

## 1.1 Challenges and Related Works

**Training and inferencing smaller capacity models in 8-bit**: A key challenge in (1-5-2) FP8 training is the ability to train smaller capacity models without losing accuracy. While networks such as VGG [18] and ResNet [10] can be trained effectively, models like MobileNetV2 that have 1/7th the capacity of ResNet50 suffer significant degradation ($\sim 1\%$) if trained in FP8, as shown in Fig.1(a). This training problem is further exacerbated when we reduce the layer width of MobileNetV2 by 0.5 and 0.35—resulting in $2 \sim 3\%$ degradation. Furthermore, as discussed in the previous section, inferencing on the post-training quantized MobileNetV2 models [7] using INT8 and INT4 formats results in significant accuracy degradation (>2%). Recent work has identified the small variance of the depthwise convolution layers as the cause of such degradation [14], which has been partly addressed in [15] by retraining the networks with the adaptable dynamic range. Techniques that can resolve this low-precision training challenge for smaller capacity models and simultaneously avoid the issues in post-training quantization can be extremely important for Edge deployment use cases.

**Applicability of 8-bit Training to Other Domains**: In Natural Language Processing(NLP) and Speech domains, popular networks built on LSTMs and Transformer blocks perform simple matrix multiplications using fully-connected (FC) layers rather than convolution operations. Training these networks using FP8 has proven to be a significant challenge. As an example, we've observed slow convergence and lower BLEU scores on the Transformer model on the WMT14 dataset trained using (1-5-2) FP8 precision, as shown in Fig.1(b). Furthermore, in many state of the art Speech and language models, the last FC layer has a very large dimension—corresponding to vocabulary size(typically 10-100 times larger than ImageNet) [19, 20]. As a result, the last FC layer consumes a significant fraction (> 20-30%) of the total computation time. Currently, 8-bit training solutions

customized for convolution nets relax the last layer precision by keeping that layer in 16-bit (FP16) since last layers computed in FP8 have shown to increase classification error rates [2]. However, this solution is expensive for NLP and Speech tasks which have large last layers. In addition, Object Detection and semantic segmentation networks such as MaskRCNN [21] and SSD [22] that load a pre-trained ImageNet backbone model and fine-tune it with an Object Detection dataset have not been investigated within the framework of 8-bit training. Finally, Trans-Precision Inference in (1-5-2) FP8 (directly converted from FP32 trained models) results in significant accuracy degradation in many of these models as shown in the table of Fig.1(c). The goal of this paper is to enable an 8-bit training methodology that addresses all of the above challenges.

**8-bit weight updates**: The weight update phase of low precision training requires a master copy of weights in higher precision to accumulate gradients across minibatches without losing critical information due to "swamping" [23]. In INT 8 training, FP32 is used for this master copy, resulting in increased latency due to the bandwidth needed for communication and AXPY ($Y = AX + Y$) computation in 32-bits. In FP8 training, even with stochastic rounding techniques, 16-bit (FP16) weights are still needed for the master copy to preserve convergence [2]. As a solution to this problem, weight averaging has been proposed to facilitate exchange of 8-bit weights (while keeping higher precision weights locally). This scheme, however, results in >4% accuracy degradation on ResNet18/ImageNet [24]. An ideal weight update scheme should compress gradients and only compute and communicate 8-bit weights during the training process.

## 1.2 Contributions

In this paper, we introduce a new **hybrid FP8** format and technique that is applicable to both computations (training and inference) and communication to address all of these challenges. In comparison to the state-of-the-art FP8 training and INT8 inference solutions, our primary contributions include:

1. A novel hybrid FP8 format that uses **4 exponent bits and 3 mantissa bits** (1-4-3 with an exponent bias) for forward propagation and 5 exponent bits and 2 mantissa bits (1-5-2) for backward propagation—achieving negligible accuracy degradation on previously problematic models including MobileNetV2 and Transformer.

2. Demonstrated the robustness of the HFP8 format on a wide spectrum of DNN tasks including Image Classification, Object Detection, NLP and Speech—while fully preserving accuracy.

3. Through theoretical analysis, we've identified BN statistics as the primary reason for accuracy loss in low-precision Trans-Precision inference and show that BN statistics could be fine tuned to fully recover model accuracy while using our 1-4-3 FP8 precision.

4. Introduced a deterministic FP8 weight update scheme that can converge to baseline accuracies without using stochastic rounding along with a compatible all-reduce technique that takes advantage of low bit-width weights to speed up distributed learning.

## 2 New Hybrid FP8 (HFP8) Formats and Computations

### 2.1 Impact of FP8 formats on Trans-Precision Inference (Post-Training Quantization)

In this section, we explore how different FP8 precision formats for activations and weights impact Trans-Precision Inference accuracy. Towards that end, we adapt the theoretical framework of Sakr et al. [25] to quantify the **mismatch probability** between a reduced precision neural network and its full-precision counterpart. Consider a neural network for a classification task such as MobileNetV2, with quantized weights ($W + q_w$) and activations ($A + q_A$), where each numerical output ($Z_i$) after the feedforward pass may be corrupted by a quantization noise ($q_{zi}$). Using Taylor's expansion and ignoring the cross-products of quantization noise terms, the total quantization noise $q_{z_i}$ can be expressed as [25]:

$$q_{z_i} = \sum_{a_h \in \mathcal{A}} q_{a_h} \frac{\partial z_i}{\partial a_h} + \sum_{w_h \in \mathcal{W}} q_{w_h} \frac{\partial z_i}{\partial w_h}, \tag{1}$$

where $\mathcal{A}$ and $\mathcal{W}$ are index sets. By evaluating the probability of any pair of outputs ($z_i < z_j$) that flipped due to quantization errors $P_r(z_i + q_{z_i} > z_j + q_{z_j})$, the mismatch probability $p_m$ between the reduced precision network and its full precision baseline yields an upper bound—defined by the

quantization error of each activation and weight multiplied by the corresponding gradients called "gain". As the gains are network specific, we can evaluate them empirically using Eqn.1.

Fig.2 shows the computed mismatch probability due to activation and weight quantizations for each layer of the MobileNetV2 (CIFAR-10) model. The results clearly show that by moving just one bit from the exponent (1-5-2) to the mantissa (1-4-3), the mismatch probability corresponding to both activations and weights decrease dramatically. This improvement comes from the fact that weights and activations are represented with higher fidelity using the extra mantissa bit.

However, since the total bit-width is limited to 8, reduction in the exponent bit-width can result in clamping of large weights and activations and/or truncation of small values to the minimum representable value in (1-4-3). Given the typical numerical distribution of these tensors during training, we found that underflow represents a more serious concern. To mitigate this effect, we introduce a fixed exponent bias that shifts the coverage range of the (1-4-3) FP8 format to $[2^{-2^{ebit-1}-bias+1}, \frac{2^{mbit+1}-1}{2^{mbit}} \times 2^{2^{ebit-1}-bias}]$.

By choosing an exponent bias of 4, we intend to better align the (1-4-3) format with the distributions of activations and weights seen in a wide range of DNN layers and models. As verified in Fig.2, introducing an extra bias of 4 on the exponent further reduces the impact of quantization—specifically on the weights in the lower layers which appear to have much smaller magnitudes. In the (1-4-3) FP8 with bias=4 format, we reduce the maximum clamping value from 480 down to 30, large enough to cover the wide variety of networks that we have investigated. In exchange, we are able to represent smaller activations and weights down to $2^{-11}$ (much lower than the $2^{-7}$ truncation threshold in 1-4-3). For simplicity of notation, all the following (1-4-3) FP8 experiments have a default exponent bias of 4. These experiments indicate that the 5-bit exponent range $2^{-15} - 2^{16}$ is an overkill for DNN inference, and 4 bit exponents with a bias of 4 have sufficient range and fidelity to represent activations and weights for both training and Trans-Precision inference performance. Finally, we've verified that these conclusions extend to a large number of neural network models and topologies.

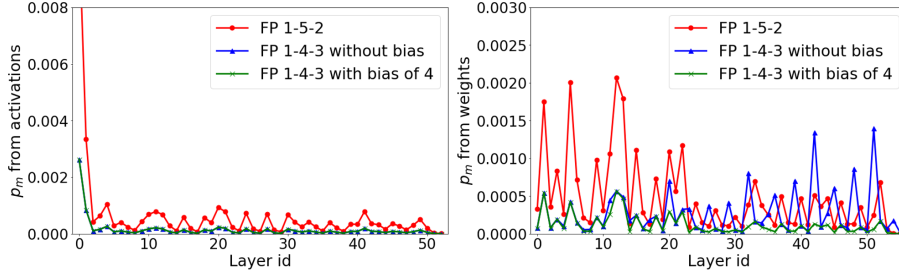

Figure 2: The layer-wise decomposition of mismatch probability (Eqn.1) for (a) activation and (b) weight quantizations on a MobileNetV2 (CIFAR-10) model (excluding the first and last layers which are in full precision). (1-5-2) results in higher activation errors compared to (1-4-3) with or without bias=4. (1-4-3) with bias=4 shows the lowest mismatch thanks to the extra fidelity needed for representing small weight values near the network output.

## 2.2 Impact of FP8 Formats on Model Training Accuracy

In addition to increasing mismatch probability, we note that quantization noise also degrades the Lipschitz property of loss surfaces, that is, the loss changes in a faster rate, and the magnitudes of the gradients are larger too. In Fig.3, we plot (a) the loss surfaces of a FP32 trained model and (b) a (1-5-2) FP8 trained model along two random directions with their coefficients scanned along the x and y axis [26]. The loss surface of the (1-5-2) trained model shows multiple saddle points and appears rougher—making gradient descent based training unstable as evidenced by the kinks in Fig.3(b). The mitigation of such kinks has also explained the effectiveness of Batch Normalization [27]. In contrast, by increasing the number of mantissa bits from 2 to 3 for the forward pass only (while keeping gradients and errors in 1-5-2), the loss-surface appears to be significantly improved in Fig.3(c), implying easier optimization. On the other hand, comparing the loss surfaces for training and test, we can see that FP8 quantizations do not impact generalization.

Guided by these insights, we propose our Hybrid FP8 (HFP8) formats utilizing two different FP8 formats to customize the precision separately for the forward and backward passes of DNN training—

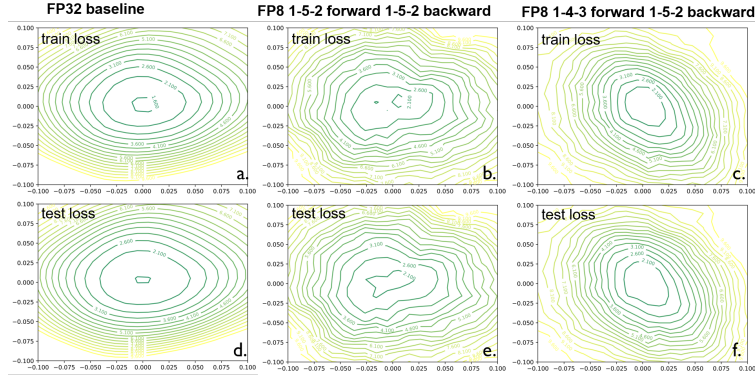

Figure 3: The loss surfaces of models trained in different precisions: (a) FP32, (b) FP8 (all GEMMs in 1-5-2, [2]), (c) HFP8 (1-4-3 only for forward pass). The top row for the loss surfaces from training data while the bottom row from test data. The loss surfaces with HFP8 maintain good Lipschitz properties compared to FP32 while the loss surfaces with FP8 exhibit multiple saddle points which hinder training convergence.

improving the performance on training and Trans-Precision inference. The underlying reason for this choice is that forward and backward passes have different optimal balances between range and precision. While tensors in the forward pass prefer higher precision (and lower representational error), gradients in the backward pass prefer a higher dynamic range. We describe our HFP8 training methodology where weights and activations adopt the (1-4-3) format (bias=4) while tensors used in backpropagation continue to be represented using the (1-5-2) format (in combination with loss scaling techniques pioneered by [28])(see Fig.1 in Appendix A). Our experiment shows that this simple change can significantly improve the performance of both MobileNet and Transformer models (as shown in Fig.4(a) and (b)), in comparison to forward (1-5-2) based results that showed significant degradation in Fig.1. In the following sections, we will show that this improvement is universally applicable to both training and Trans-Precision inference (training results for Speech and Object Detection models are shown in Fig.4(c) and (d)).

For errors and gradients in HFP8 back-propagation, we employ the (1-5-2) FP8 format, which has proven to be optimal across various deep learning tasks. However, unlike activations and weights, even 5-bit exponents are insufficient to represent the wide dynamic range seen in activation gradients. Therefore, loss scaling has been adopted to enable gradients to become large enough to be representable using the (1-5-2) format [2]. Nonetheless, it's infeasible to seek a unique scaling factor that fits a wide range of different models and datasets. Towards that end, we adapted auto-adjusted scale factors for gradients and errors during HFP8 training using Apex [28] (details in Appendix B). Finally, through hardware design experiments, we've confirmed that floating-point units (FPUs) that can support both formats are only 5% larger than the original FPUs that only support 1-5-2.

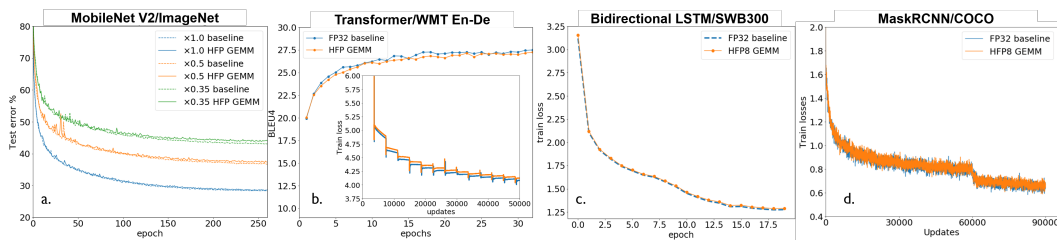

Figure 4: Training curves using HFP8 on (a) MobileNetV2 with different width-multipliers (sizes) (b) Transformer-base machine translation (c) LSTM-based Speech Model for the SWB300 dataset and (d) Mask R-CNN model. No significant loss in accuracy is observed across DNN layer types, models and datasets. Final training results on a diverse set of models are summarized in Table4.

## 2.3    Last Layer Precision and the SoftMax Function

For networks with large output dimensions (typically seen in Speech and NLP), the last FC and SoftMax layers contribute to a significant fraction of the total computation time due to large matrix-multiplications and expensive exponential functions (especially if these need to be computed in FP16). In these layers, it therefore becomes critical to be able to use 8-bit computations.

First, we note that when (1-4-3) FP8 is used along with (1-6-9) FP16 output precision no degradation on LSTM-based SWB300 and Transformer-based translation tasks is observed. In contrast, when the output precision of the FC layer is set to (1-4-3) as well, large loss in accuracy is observed (network diverges in SWB300 and $\sim 1$ BLEU degradation in WMT En-De). This occurs because the largest output of the last FC layer may be quantized into the same values (bins) during conversion from 16 to 8-bit and therefore become indistinguishable to the ensuing SoftMax layer. Fig.5(a) and (b) shows the distribution of output activations before and after FP8 (1-4-3) quantization in the transformer model for WMT En-De translation($d_{out} = 42720$), showing that the largest numbers are poorly represented by 8-bit in Fig.5(b). Interestingly, we discovered that if the quantization step is performed after the max subtraction step (i.e. $x - x_{max}$) in SoftMax, this degradation in accuracy can be fully eliminated. In Fig.5(c), the $x - x_{max}$ sub-step of SoftMax moves the largest values closest to 0, where data representation is strongest due to non-uniform nature of floating point representation. Furthermore, this technique also allows SoftMax to be performed using just 8-bits. Detailed discussions on the reduced precision SoftMax will be a focus of future work. Overall, the (1-4-3) HFP8 format in the last FC layer when combined with an output precision of 1-6-9 and the max-subtracted SoftMax function allows for efficient end-to-end HFP8 computations.

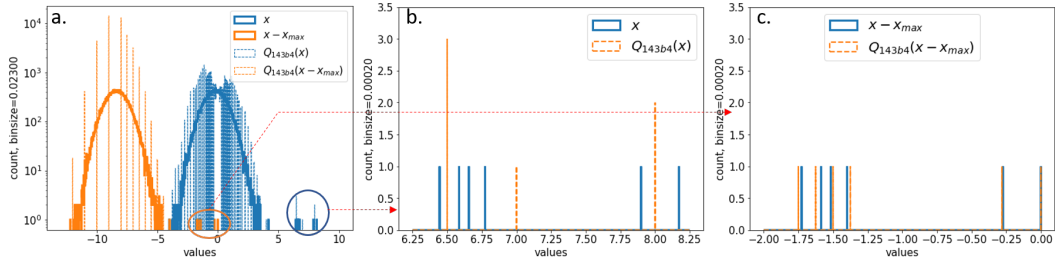

Figure 5: Output activation distributions of the last FC layer and its quantization (to 1-4-3 FP8) in Transformer model before and after subtracting the max output. Quantization after subtracting with the max output allows the largest inputs of SoftMax to be represented with high fidelity in 8-bits and fully preserves model accuracy.

## 3    Trans-Precision Inference in FP8

Guided by the theoretical framework in Section 2.1, we investigate how inference accuracies are impacted when FP32 trained models are used directly with different FP8 formats for inference (i.e. without any retraining). Using MobileNetV2 trained on ImageNet as an example, we immediately observe that the (1-4-3) FP8 format is significantly more accurate than the (1-5-2) format—as shown in the first 2 rows of the Table in Fig.6(a). This is consistent with mismatch probability based predictions described earlier. However, even with the right FP8 format, we observe that we lose >5% in model accuracy in comparison to FP32. To reduce this gap, we provide 2 additional insights. The first key insight comes from a theoretical understanding of how quantization errors in weights and activations directly impact the accuracy of outputs of the succeeding Batch Normalization (BN) layer. Retuning the statistics (mean and variance) of the BN layer for the precision of interest (i.e. inference precision) has the potential to significantly reduce this error. As shown in Eqn.2, the quantization error at the output of a BN layer ($Z - Z_Q$) can be expressed in terms of the variance of quantization error in BN input $\sigma_Q^2$ and the variance of precise input $\sigma_Y^2$—assuming $Q$ and $Y$ are not correlated (please see Appendix C for a detailed derivation):

$$E[\|Z - Z_Q\|_2] \begin{cases} \cong \gamma^2 \frac{\sigma_Q^2}{\sigma_Y^2}, & \text{original BN statistics.} \\ \cong 2\gamma^2(1 - \frac{1}{\sqrt{1 + \frac{\sigma_Q^2}{\sigma_Y^2}}}), & \text{retuning BN statistics.} \end{cases} \tag{2}$$

Table 1: FP8 **Trans-Precision inference** for FP32 trained models after BN re-tuning (if applicable)

| Model (Dataset) | Baseline (FP32) | **1-4-3 Inference** | 1-5-2 Inference |
|---|---|---|---|
| *ResNet18* (ImageNet) | 69.32 | **68.99** | 68.93 |
| *ResNet50* (ImageNet) | 76.44 | **76.46** | 75.89 |
| *DenseNet121* (ImageNet) | 74.76 | **74.78** | 74.40 |
| *AlexNet* (ImageNet) | 57.10 | **57.07** | 56.87 |
| *MobileNetV2* (ImageNet) | 71.81 | **71.37** | 70.31 |
| *4-bidirectional-LSTM Speech* (SWB300)[a] | 9.90 | **9.90** | 10.10 |
| *Transformer-base* (WMT14 En-De)[b] | 27.53 | **27.47** | 27.06 |
| *SSD-Lite* (VOC)[c] | 68.79 | **68.22** | 67.40 |
| *MaskRCNN* (COCO)[d] | 33.58/29.27 | **33.43/29.10** | 32.83/28.65 |

[a]Word Error Rate[b]BLEU score[c] mean average precision(mAP)[d] Box/Mask average precision

Plotting this equation in Fig.6(b), we observe that $E[\|Z - Z_q\|_2]$ increases linearly with $\sigma_Q^2$ when preserving original BN parameters, but increases only sub-linearly when BN statistics are re-tuned. This reveals the fact that the impact of quantization at BN output would be suppressed once BN statistics are properly tuned. As shown in Rows 5 and 6 of Fig.6(a), re-tuning BN statistics using just 2% of a single epoch of the training dataset reduces this accuracy gap significantly.

The second key insight obtained using Eqn.2 indicates that layers that have very low $\sigma_Y^2$ have vastly magnified output errors. Plotting $\sigma_Y^2$ as a function of layer number for MobileNetV2 (Fig.6(c)) leads us to note that depthwise (DW) convolution layers produce activations that have orders of magnitude smaller variance ($\sigma^2$) in comparison to traditional convolution layers [14]. We therefore expect the precision setting in these layers to strongly impact Trans-Precision inference accuracies. Since DW layers contribute to <3% of the overall compute in the MobileNet family of networks [17], we recommend setting the precision in these layers uniformly to FP16. As can be seen from Rows 3,4 (without BN re-tuning) and Rows 7,8 (with BN re-tuning), this precision setting in the DW layers substantially improves inference accuracies to within $\sim 1\%$ of the baseline.

A combination of these techniques – (a) picking the right precision format (1-4-3) for weights and activations of convolution and FC layers (b) setting the precision for DW layers to FP16 and (c) updating BN $\mu$ and $\sigma^2$ with minimal training data – allows MobileNetV2 to hit accuracies within $\sim 0.5\%$ of the full precision baseline. Furthermore, we show that these techniques extend very well to other models; as shown in Table 1, FP8 1-4-3 with BN re-tuning can fully recover the baseline inference accuracies for the entire spectrum of networks studied. Note that for BN re-tuning, data does not need to be labeled and thus can be done at the edge devices.

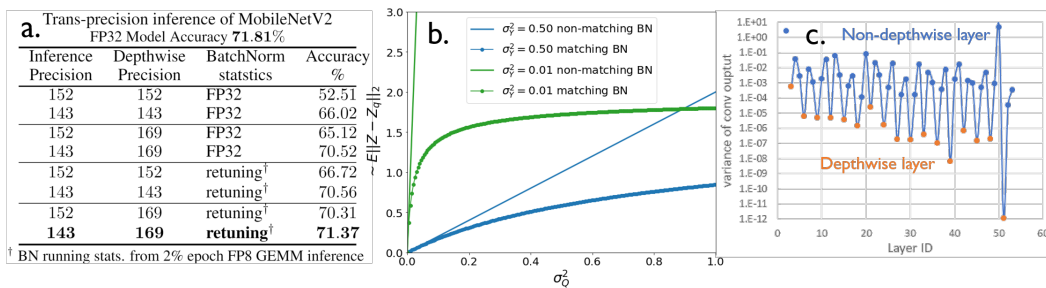

Figure 6: (a) Trans-Precision inference accuracies using the FP32 MobileNetV2 Model in two different FP8 formats. Rows 5-8 further re-tune BN $\mu$ and $\sigma^2$ using 2% of the training data while using FP8 precision. (b) Visualization of the quantization error (Eqn.2) at BN output vs. the quantization error variance at BN input for two different variances of precise BN input (c) The variances of precise BN input for 52 convolution layers in MobileNetV2 showing DW layers having orders of magnitude smaller $\sigma^2$.

## 4 Hybrid FP8 Distributed Training Scheme and Results

As DNN compute functions are accelerated in each learner using HFP8, the communication cost among learners and memory bandwidth dominated tasks like weight updates become bottlenecks.

Table 2: Ring based communication schemes among $N$ learners using Hybrid FP8

| Common ring-based weight update | Proposed ring-based weight update | & data format sent |
|---|---|---|
| (1) reduce-scatter gradients | (1) reduce-scatter gradients | FP16(1-6-9) |
| (2) all-gather gradients | (2) locally update $1/N$ gradients to weights | local |
| (3) locally update full size of weights | (3) all-gather weights | FP8 (1-4-3) |

Table 3: Round-off residual based Hybrid FP8 weight update(per worker)

| For each 1/N weight: $R_{t=0} \leftarrow 0$ | (Initialize round off residual) |
|---|---|
| For timestep $t$ for each 1/N weight: | |
| $\hat{W}_t \leftarrow W_{t-1} - \alpha_{t-1}\mathbf{g}(W_{t-1}) - R_{t-1}$: | (Apply gradients and carried-on residuals) |
| $W_t \leftarrow \mathbf{Q_W}(\hat{W}_t)$ | (Quantize new weights) |
| $\hat{R}_t \leftarrow W_t - \hat{W}_t$ | (Overwrite residuals) |
| $R_t \leftarrow \mathbf{Q_R}(\hat{R}_t)$ | (Quantize residuals, higher precision than $\mathbf{Q_W}$) |

Table 4: Baseline vs. Hybrid FP8 training on Image, Language, Speech and Object-Detection Models

| Model(Dataset) Accuracy or [other metrics] | Baseline(FP32) | HFP8 + Round-off update |
|---|---|---|
| *AlexNet* (ImageNet) | 57.28 | 57.21 |
| *ResNet18* (ImageNet) | 69.38 | 69.39 |
| *ResNet50* (ImageNet) | 76.44 | 76.22 |
| *MobileNetV2* (ImageNet) | 71.81 | 71.61 |
| *DenseNet121* (ImageNet) | 74.76 | 74.65 |
| *2-LSTM* (PennTreeBank)[Test ppl.] | 83.66 | 83.86 |
| *Transformer-base* (WMT14 En-De)[BLEU] | 27.50 | 27.27 |
| *4-bidirectional-LSTM Speech* (SWB300)[WER] | 9.90 | 10.00 |
| *MaskRCNN(ResNet50)* (COCO)[Box/Mask AP] | 33.58/29.27 | 33.06/28.86 |
| *SSD-Lite(MobileNetV2)* (VOC)[mAP] | 68.79 | 68.72 |

Hardware performance estimations indicate that this communication could take up to $\sim 41 - 62\%$ of the end-to-end training time for ResNet50 with HFP8 GEMM (Appendix D for details). As illustrated in Table 2, the conventional communication pattern used in deep learning algorithms exchanges gradients through ring-based all-reduce [29, 30] and then each learner updates the whole model locally. To take advantage of 8-bit weights in off-chip communication as well as to minimize local memory bandwidth, we modify the existing distributed learning scheme slightly—so that each of $N$ learners updates only $1/N^{th}$ of the model after the reduce-scatter phase minimizing local memory transactions. When updating the model globally, the final 8-bit weights produced in each learner are distributed in the all-gather phase, thereby improving off-chip communication by $2\times$ compared to conventional 16-bit gradient communication.

To improve the robustness of low-precision weight updates and to prevent "swamping" [2], we propose a deterministic round-off residual update scheme that stores the weight in 8-bit while saving the quantization errors locally as "round-off" residuals in FP16 as illustrated in Table 3. We study this round-off residual scheme on a wide range of DNN applications and show that it does not impact convergence (consistent with the rich body of theoretical work in this space [31, 32]). With 8-bit weight updates and a modified ring-distribution scheme, our technique improves end-to-end training time by $32 - 38\%$ on ResNet50 (for details, see Appendix D).

Finally, to demonstrate the wide applicability and the robustness of the HFP8 formats, 8-bit computations and round-off residual scheme, we tested it on a wide spectrum of deep learning models and datasets without changes to network architectures, data pre-processing, or hyper-parameters(details in Appendix E). **As shown in Table 4, every single network tested achieved accuracy very close to the full precision baseline**, including tasks that were problematic for previous FP8 endeavors (such as MobileNet and Transformers). More complex and challenging tasks, such as Object Detection, Speech and Machine Translation in HFP8 are demonstrated and for the first time show performance within $0.5\%$ of the full precision baseline on large networks and datasets. Given the limited computational complexity in the first and last layers we set the precision in these layers to FP16 except for Speech and Transformer networks, where we use the same HFP8 settings on the large final FC layer and find no degradation.

# 5 Conclusions

We have demonstrated DNN training with a new Hybrid FP8 format that adopts two different FP8 formats for forward and backward propagation. In addition, we introduced a novel round-off residual scheme which can significantly improve robustness of low-precision AXPY and reduce communication bandwidth requirements. We've confirmed the superior accuracy of this approach over previous 8-bit training proposals on a wide range of state of the art DNN models. In addition, we've presented new insights in Batch Normalization and depthwise Convolutional layers that demonstrate how the same FP8 format can be used for highly accurate Trans-Precision inference (starting from higher precision FP32 models). These novel techniques enable a new generation of 8-bit hardware systems that are robust for the entire spectrum of DNN training and inference applications.

**Acknowledgments**

The authors would like to thank Lam Nguyen and Charbel Sakr for helpful theoretical discussions, Anthony Giordano, I-Hsin Chung, Ming-Hung Chen, Paul Crumley, Kaoutar El maghraoui, and Jeffrey Burns for the computing infrastructure, and Leland Chang, Sunil Shukla, Silvia Mueller, Ankur Agrawal, Jinwook Oh, Marcel Schaal, Mauricio Serrano, Wei Wang and the team for the chip platform targeted in this work. This research is realized by generous collaborations across IBM Research.

## Footnotes

*contributed to this work while at IBM, is currently with the Electrical Engineering Department, Hanyang University, South Korea, email: choij@hanyang.ac.kr

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
