[Supplementary Material · Supplemental_Hybrid_FP8_NeurIPS19_camera_ready.pdf]

# 1 Appendix A Details about Proposed Hybrid 8-bit Floating Point Format

Figure 1: The detailed Hybrid FP8 representation and computation precisions proposed for the forward, backward, gradient computation and weight update phases of DNN training. The proposed HFP8 scheme allows us to preserve accuracy for every DNN model explored in this study and enables significant end-to-end training system-level performance improvements

For the first and the last layer, unless specified, the data and Fused-Multiply-Add(FMA) precisions are FP16 1-6-9. The rest of the layers are referred to as "middle layers".

## 1.1 Forward GEMM

For the middle layers, weights and activations are used in the format of FP8 1-4-3 with bias=4. In the forward pass, the accumulation precision of FMA is FP16 1-6-9 format. The output precision is FP16 1-6-9. The FP16 data will enter ReLU, BatchNorm and special function layers and be quantized into FP8 1-4-3 again in the next Convolution(Conv) or Linear layer for GEMM operations.

## 1.2 Backward GEMM

For the middle layers, activation gradients will be in FP8 1-5-2 format. In the backward pass, the accumulation precision of FMA is FP16 1-6-9. Backpropagation uses 1-5-2 activation gradients and 1-4-3 weights with the output precision of 1-6-9. After the backward pass of the special functions, the output is quantized to 1-5-2 again for the previous FC or Conv layers.

## 1.3 Weight gradient(WGRAD) GEMM

For the middle layers, WGRAD calculation does FMA on 1-4-3 activations and 1-5-2 activation gradients, then, outputs 1-6-9 WGRAD. The accumulation precision of GEMM is FP16 1-6-9 format.

## 1.4 Weight update

For FP8 round-off weight update, the weight is in FP8 1-4-3 format, WGRAD is in FP16 1-6-9 format, the momentum and other intermediates are also in FP16 1-6-9. The output precision of the weight update is FP8 1-4-3. The residual is quantized into FP16 1-6-9.

## 1.5 GEMM accumulation

To avoid "swamping"[1], we adopt a two-level chunk-based accumulation in GEMM operations with the chunk size of 64. For two N-dim vectors' dot product, 64 pairs of FP8 operands are multiplied and accumulated with the partial sum precision in FP16 1-6-9. Then $N/64$ partial sums are accumulated for the final product in FP16 1-6-9.

# 2 Appendix B: Experimental setup

## 2.1 Quantization-enabled PyTorch Framework

Our experiments were performed on Nvidia V100 GPUs. Due to the absense of HFP8 hardware, we used CUDA C++ function to emulate the process of quantization. The numbers are clamped to maximum or minimum allowed values of the chosen format if overflow or underflow occurs. Then they are quantized by truncating the mantissa bits with nearest-rounding. For special quantization, such as stochastic rounding, CuRAND function is used for generating randomness at the levels of each thread and each call of the quantization code.

To realize flexible precision control of data, GEMM and weight update, we modified PyTorch code base from the CUDA C++ *.cu files at the bottom of the hiearchy to the python nn.modules at the top. In particular, we added new tensor methods like simple matrix-multiply for the FC layers and batched-matrix-multiply for the Transformer network. For different types of convolutions, we added new CUNN native functions for the common, depthwise and upsampling(transpose) convolutions. We also added special functions like SoftMax and Pointwise quantization. All the new methods and functions are compatible with the auto-grad mechanism of PyTorch. In the new methods and functions, we kept the data manipulation codes unchanged, such as tensor lowering and raise, and took over the process right before calling cuBLAS functions like *Sgemm()*. Instead of using cuBLAS, which is not open-sourced, we called our own customized CUDA functions to implement quantization at the single FMA level.

We connected these methods up to the python nn.modules, so that the model files can directly call the quantized FC, Conv2d, depthwise-Conv2d and other modules to build up a network and pass down precisions to the CUDA codes. For the weight update, we wrote new Optimizers(a PyTorch module to update weight) with input arguments including precisions and exclude lists, so that we could decide the low precision weight update scheme and the layers to quantize. This solution requires almost no revisions except adding precision conditions to the original run files, giving us convenience to quickly study the low precision training of any public available deep learning networks.

Within data precision, we can control the precision for the forward, backward and WGRAD computation. Within FMA precision, we can control the partial sum precision and the final output precision. We verified the robustness of the implementation in PyTorch with unit tests. Fig.2,3,4 show a comparison between using the original FP32 Conv2D layer and using the new quantizing Conv2d**R** layer, which performs forward, backward, and WGRAD in any desired precisions. In these examples, we deliberately used low output precision to show the quantization effect. An example of the built-up network is shown in Fig.5, in which a ResNet is constituted of the first Conv layer in 1-6-9 precision and the following Conv layer in 1-4-3 forward and 1-5-2 backward precisions. The WGRAD input precisions are determined by the activation precision 1-4-3 and the activation gradient precision 1-5-2, where the output precision is 1-6-9 same as FMA precision(acc_prec).

```
 1  # Quantized Conv2d named as Conv2dR, taking precision args. in [data, FMA]
 2  # make FMA_prec = data_prec to show the quantization effects
 3  conv = nn.Conv2d(1,1,(3,3)).cuda()
 4  convR = nn.Conv2dR(1,1,kernel_size=3,
 5      mantissa=[4,4], backward_mantissa=[3,3], acc_mantissa=[2,2] ).cuda()
 6  #assign weights, conv_xxx = convR_ xxx
 7  conv.weight, conv.bias = Parameter(conv_filter), Parameter(conv_bias)
 8  convR.weight, convR.bias = Parameter(convR_filter), Parameter(convR_bias)
 9  #Forward e.g. mbit=4
10  #convR_in = conv_in = 5x5 random numbers
11  conv_out = conv(conv_in)
12  convR_out = convR(convR_in)
13  print("FP32 forward: {}".format(conv_out))
14  print("\nquantized forward: {}".format(convR_out))
```

```
FP32 forward: tensor([[[[-2.5105, -1.8337, -1.6522],
          [ 0.8606,  4.1457,  0.0606],
          [ 0.8449,  0.7014, -0.0225]]]],
       device='cuda:0', grad_fn=<ThnnConv2DBackward>)

quantized forward: tensor([[[[-2.6250, -1.7500, -1.5625],
          [ 0.8125,  4.2500,  0.0625],
          [ 0.8125,  0.6875,  0.0000]]]],
       device='cuda:0', grad_fn=<ThnnConv2DBackward>)
```

Figure 2: A simple convolution forward, FP32 vs mbit=4. The Quantizing Conv function is Conv2dR taking precisions(highlighted lines) and also calling a quantizing grad_fn in the backward pass(highlighted in the output)

```
 1  #Backward eg. mbit=3 for simplicity just added up the loss as a scalar of error to back propergate
 2  conv_loss = torch.sum(conv_out.view(-1))
 3  conv_loss.backward()
 4  conv_loss_R = torch.sum(convR_out.view(-1))
 5  conv_loss_R.backward()
 6  print("FP32 dL/dX: {}".format(conv_in.grad))
 7  print("\nquantized dL/dX: {}".format(convR_in.grad))
```

```
FP32 dL/dX: tensor([[[[ 0.3386, -0.5615, -1.3742, -1.7129, -0.8128],
          [-0.1915,  0.1654, -0.2646, -0.0731, -0.4300],
          [ 0.1154,  1.3807,  0.1537,  0.0383, -1.2271],
          [-0.2233,  1.9422,  1.5279,  1.7512, -0.4143],
          [ 0.3069,  1.2154,  0.4183,  0.1114, -0.7971]]]], device='cuda:0')

quantized dL/dX: tensor([[[[ 0.3438, -0.6250, -1.3750, -1.7500, -0.8125],
          [-0.2188,  0.0938, -0.2812, -0.1250, -0.4375],
          [ 0.0938,  1.5000,  0.0938,  0.0000, -1.2500],
          [-0.2500,  2.0000,  1.5000,  1.7500, -0.4375],
          [ 0.3125,  1.2500,  0.4375,  0.1250, -0.8125]]]], device='cuda:0')
```

Figure 3: A simple convolution backward, FP32 vs mbit=3

```
 1  #Wgrad eg. mbit=2
 2  print("FP32 dL/dW: {}".format(conv.weight.grad))
 3  print("\nquantized dL/dW: {}".format(convR.weight.grad))
 4
```

```
FP32 dL/dW: tensor([[[[-3.1420, -3.1046, -1.5566],
          [ 1.5958,  1.4640, -0.7524],
          [ 3.7715,  4.0120,  2.2402]]]], device='cuda:0')

quantized dL/dW: tensor([[[[-3.0000, -3.5000, -1.5000],
          [ 1.2500,  1.5000, -0.8750],
          [ 4.0000,  3.5000,  2.5000]]]], device='cuda:0')
```

Figure 4: A simple convolution WGRAD, FP32 vs mbit=2

```
420 => Model : ResNet(
421   (conv1): Conv2dR(3, 64, kernel_size=(7, 7), stride=(2, 2), padding=(3, 3),
      bias=False | mantissa=[9, 9], exponent=[6, 6], backward_mantissa=[9, 9], ba
      ckward_exponent=[6, 6], acc_mantissa=[9, 9], acc_exponent=[6, 6])
422   (bn1): BatchNorm2d(64, eps=1e-05, momentum=0.1, affine=True, track_running
      _stats=True)
423   (relu): ReLU(inplace)
424   (maxpool): MaxPool2d(kernel_size=3, stride=2, padding=1, dilation=1, ceil_
      mode=False)
425   (layer1): Sequential(
426     (0): BasicBlock(
427       (conv1): Conv2dR(64, 64, kernel_size=(3, 3), stride=(1, 1), padding=(1
      , 1), bias=False | mantissa=[3, 9], exponent=[4, 6], backward_mantissa=[2, 9
      ], backward_exponent=[5, 6], acc_mantissa=[9, 9], acc_exponent=[6, 6])
428       (bn1): BatchNorm2d(64, eps=1e-05, momentum=0.1, affine=True, track_run
      ning_stats=True)
```

Figure 5: *print(Model)* of ResNet 18 with the first conv layer in FP16 and the following in FP8

## 2.2 The Adaptation of Nvidia APEX package for the auto-scaling of FP8 1-5-2 activation gradients

Nvidia APEX package [2] was designed to find the best scaling factor in FP16-FP32 Mixed Precision Training [3] for activation gradients in IEEE Half Float Point(1-5-10). Without causing overflow, previously underflowed numbers could be represented by the Half format after multiplying the scaling factor. Before the AXPY operation to update weights, the enlarged activation gradients are divided by the same scale factor to be numerically equivalent to the non-APEX case. At the same time, APEX seeks FP infinity(INF) value in the activation gradient tensors to detect events of overflow. If the INF occurrence exceeds a threshold (0 for zero tolerance), APEX will skip the update of this iteration and reduce the scale factor. Meanwhile if no occurrence of INF has been detected for a window of updates, APEX will increase the scale factor.

We adapted this package in our PyTorch Platform. As our computation is performed on GPUs and the overflow in FP8 will not really trigger an INF, we manually set the overflowed value to INF in our CUDA quantization functions. And we also set up the conditions to ensure that only the overflow of activation gradients will trigger an INF event—An FP8 1-4-3 overflow during forward will not generate an INF but just the maximum number in the format.

The auto-scaling of activation gradients works smoothly with our quantizaton-enabled PyTorch framework. In Fig. 6 we show how the loss scale is being automatically adjusted until stabilizes at $2^{12}$ for FP8 1-5-2 backpropagation when training a Transformer model.

Figure 6: the evolution of the loss scale of a transformer model trained on WMT14 En-De translation job

# 3 Appendix C: Understanding Batch Normalization Statistics for Trans-Precision Inference

In this section, we provide a deeper analysis on the role of Batch Normalization statistics for Trans-Precision inference. Consider a convolution layer (CONV), $Y = W * X$, followed by a BatchNorm layer (BN), $Z = \frac{Y-\mu}{\sigma}\gamma + \beta$, where $\mu$ and $\sigma^2$ are running mean and variance of BN, and $\gamma$ and $\beta$ are the BN parameters, all provided by the pre-trained model (typically trained in FP32). We focus on the impact of quantization error in CONV caused by Trans-Precision inference to the BN output, i.e., $Z_q = \frac{Y_q-\mu'}{\sigma'}\gamma + \beta$. Since all the parameters in the network are not changed in the Trans-Precision inference, the same $\gamma$ and $\beta$ are used for $Z_q$. Whereas, $\mu'$ and $\sigma'$ represent the underlying statistical characteristic of the output of CONV, thus we consider two cases: (1) $\mu' = \mu$ and $\sigma' = \sigma$ assuming that the output statistics of quantized CONV is similar to the FP32 CONV, and (2) $\mu' = E[Y_q]$ and $\sigma'^2 = Var[Y_q]$, considering that BN statistics are tuned to capture changes in CONV output due to quantization.

We start from defining an error metric, *expected quantization error at BN output*: $E[\|Z - Z_q\|_2]$. Then, from the setting, we can derive the following optimization problem:

$$\min E[\|Z - Z_q\|_2] = \min_{\mu',\sigma'} E[\left\|(\frac{Y-\mu}{\sigma}\gamma + \beta) - (\frac{Y_q-\mu'}{\sigma'}\gamma + \beta)\right\|_2], \qquad (1)$$

where the objective is to find $\mu'$ and $\sigma'$ that can minimize the impact of quantization at BN output. Since this becomes an input activation for the next CONV, it is particularly important to suppress the quantization impact at this level.

This error term can be further elaborated given the choice of $\mu'$ and $\sigma'$.

**Case 1: $\mu' = \mu$ and $\sigma' = \sigma$.** In this case, we can simply replace $\mu'$ and $\sigma'$ with $\mu$ and $\sigma$, then the error term is reduced to the following equation:

$$E[\|Z - Z_q\|_2] = \frac{\gamma^2}{\sigma^2} E[(Y - Y_q)^2] \simeq \gamma^2 \frac{Var[Q]}{Var[Y]}, \qquad (2)$$

where $Q = Y - Y_q$ is the quantization error at CONV output and we assume $E[Q] = 0$. Note that the expected quantization error grows linear to $Var[Q]$.

**Case 2: $\mu' = E[Y_q]$ and $\sigma'^2 = Var[Y_q]$.** The first (and obvious) insight in this case is that $Y_q$ will be normalized with $\mu'$ and $\sigma'$, and thus $Z_q$ will have the same mean and variance as $Z$ (i.e., it is simple to show that $E[Z] = E[Z_q]$ and $Var[Z] = Var[Z_q]$). We can imagine that these matching statistics at the output of BN would be desirable for stable inference.

The second (and less obvious) insight can be drawn from the following derivation:

$$E[\|Z - Z_q\|_2] = \gamma^2 E[\left\|(\frac{Y}{\sigma} - \frac{Y_q}{\sigma'}) - (\frac{\mu}{\sigma} - \frac{\mu'}{\sigma'})\right\|_2] \qquad (3)$$

$$= \gamma^2 E[(\frac{Y}{\sigma} - \frac{Y_q}{\sigma'})^2 - 2(\frac{Y}{\sigma} - \frac{Y_q}{\sigma'})(\frac{\mu}{\sigma} - \frac{\mu'}{\sigma'}) + (\frac{\mu}{\sigma} - \frac{\mu'}{\sigma'})^2] \qquad (4)$$

$$= \gamma^2 \left\{ E[(\frac{Y-\mu}{\sigma})^2] + E[(\frac{Y_q-\mu'}{\sigma'})^2] - 2E[(\frac{Y-\mu}{\sigma})(\frac{Y_q-\mu'}{\sigma'})] \right\} \qquad (5)$$

$$= 2\gamma^2 \left\{ 1 - E[(\frac{Y-\mu}{\sigma})(\frac{Y_q-\mu'}{\sigma'})] \right\} \qquad (6)$$

$$= 2\gamma^2 \left\{ 1 - Corr[Y, Y_q] \right\}. \qquad (7)$$

In other words, once BN statistics are tuned to match the statistics of $Y_q$, the expected quantization error can be measured as a function of the correlation coefficient between $Y$ and $Y_q$, $Corr[Y, Y_q]$. To understand this projection from the linear error model to the correlation-based model, we can simplify the correlation coefficient term:

$$Corr[Y, Yq = Y + Q] = \frac{Cov[Y, Y + Q]}{\sqrt{Var[Y]}\sqrt{Var[Y + Q]}} \qquad (8)$$

$$= \frac{Var[Y] + Cov[Y, Q]}{\sqrt{Var[Y]}\sqrt{Var[Y] + 2Cov[Y + Q] + Var[Q]}}. \qquad (9)$$

For a simple illustration, assume that $Cov[Y, Q]$ is a constant and negligible. Then, we can derive the expected error term in a simple form:

$$E[\|Z - Z_q\|_2] \simeq 2\gamma^2 \left\{ 1 - \frac{1}{\sqrt{1 + \frac{Var[Q]}{Var[Y]}}} \right\}. \tag{10}$$

Note that the error model Eqn.10 has much slower growth than Eqn.2. In other words, once BN statistics are tuned, it is expected that BN output becomes more robust to the impact of quantization in CONV.

To demonstrate this, we plot $E[\|Z - Z_q\|_2]$ by increasing $Var[Q]$ for different choice of $Var[Y]$ in Fig.7. We observe that $E[\|Z - Z_q\|_2]$ increases linearly with $Var[Q]$ when preserving the original BN statistics, but increases only sub-linearly when BN statistics are tuned. This illustrates that the BN layer, when tuned, can mitigate quantization error related changes on the output.

Figure 7: Visualization of the quantization error (Eqn.2 and 10) at BN output with varying variance of the quantization error at BN input for two different variances of precise BN input.

## 4 Appendix D: Performance Benefits using HFP8 Precision on Distributed Peta-FLOP Training Systems

This section quantifies the improvement in system performance achieved by using the proposed HFP8 number representation and associated training strategies. To this end, we considered a distributed training system comprised of multiple accelerator chips connected using hybrid cube-mesh interconnection network topology, similar to NVIDIA DGX GPU systems. Each accelerator chip is comprised of multiple cores. The core architecture is based on [4](anonymous), where each core contains a systolic array of fused multiply-and-accumulate (FMA) engines along with a private scratchpad memory and special function units. Overall, our system contains 64 accelerator chips, delivering $\sim$4 PFLOPS FP16 peak performance. In the case of FP8/HFP8 precision, based on detailed post-layout area and power analysis at 14nm technology, we were able to pack 2$\times$ the number of compute elements under the same area and power constraints as in FP16, doubling the peak performance to $\sim$8 PFLOPS.

### 4.1 Performance Estimation Framework

To estimate system performance, we employed a systematic performance analysis framework [5](anonymous), designed for DNN accelerators. Given a system configuration (number of compute elements, cores and chips, scratchpad capacity and organization, interconnect topology and bandwidths), the framework constructs a Design Space Configuration (DSC) to analytically capture all possible ways to map DNN computations on the accelerator system. It is equipped with a bandwidth-centric, throughput oriented performance model which accurately accounts for computations executed in each core and data communicated through each interconnect link to provide

an estimate on performance. Using the DSC, it explores various optimizations such as the choice of data *vs.* model parallelism, intra- and inter-layer reuse aware datastructure placement in scratchpads, software pipelining to overlap communication with compute, and others, to identify the mapping configuration that yields the best performance.

## 4.2 Gradient Accumulation and Weight Distribution with HFP8

In distributed training systems, the communication time to reduce weight gradients across chips can take between ∼40-60% of the total runtime based on the available off-chip bandwidth and is one of the key determinants of system performance. Compared to FP16 and FP8 representations, the proposed HFP8 number representation reduces the amount of data communicated between chips, consequently improving the time for gradient reduction. We explain the key sources of benefits using HFP8 in this subsection.

Gradient reduction on distributed systems involves three key phases: (i) reduce-scatter, where weight gradients from different chips are accumulated, (ii) all-gather, where the accumulated weight gradients are broadcast to each chip, and (iii) weight-update, where each chip updates the model using the accumulated weight gradients. Both FP16 and FP8 exchange gradients in 16 bits during both reduce-scatter and all-gather phases. Also, each chip updates the entire model using the accumulated gradients. In contrast, with HFP8 the all-gather and weight-update phases are modified as follows. The weight-update phase occurs first, where each chip updates only a portion of the model for which it has the accumulated gradients. By maintaining a residual vector for the weights that the chip updates, they are computed in 8 bits. Then, in the all-gather phase, the updated weights from each chip is broadcast to all other chips in 8 bits. Thus HFP8 improves the performance of both all-gather and weight-update phases.

We note that HFP8 is complementary to gradient compression methods such as [8][9]. By compressing weight gradients, they improve the reduce-scatter phase, whereas HFP8 focuses on the other two phases.

## 4.3 Training Throughput Improvement with HFP8

Figure 8 presents the improvement in training throughput achieved using FP8 [6] [7] and HFP8 (this work) number representations compared to FP16 for the ResNet50 benchmark. Training throughput is quantified as the number of training images processed in the overall system per second (in units of thousands). We observe from Figure 8 that going from FP16 to FP8 yields ∼31% improvement in throughput. Although the number of peak FLOPs is doubled compared to 16 *vs.* 8 bit GEMM operation, the throughput improvement at the system-level is limited primarily by the time its takes for off-chip communication to reduce gradients and update weights which still happens at 16 bit precision. The HFP8 number representation reduces off-chip communication as weights are communicated between chips using 8 bits, which provides an additional 28% boost in system throughput (detailed analysis in Section 4.4).

Figure 8: Improvement in training throughput for ResNet50 using FP16, FP8 and HFP8 number representations

We also quantified the benefits of HFP8 in conjunction with previously proposed gradient compression techniques such as AdaComp [8] and Deep Gradient Compression [9]. We considered various gradient

compression factors starting from $32\times$ to $512\times$. As the compression factor grows, the fraction of execution time spent in off-communication is reduced. As a result, the gap between FP16 and FP8 is enhanced up to 38%. Similarly, the benefits of HFP8 over FP8 is also more pronounced, as weight update and broadcast becomes the key determinant of the system performance with gradient compression.

## 4.4 Off-chip Communication Breakdown Analysis

To provide more insights into the source of benefits between FP8 and HFP8, Figure 9 breaks down the communication time into reduce-scatter, all-gather and weight-update phases for different off-chip bandwidth choices and demonstrates how each one of them is impacted by using HFP8.

Figure 9: Analysis of communication time for gradient reduction and weight distribution

First, at low off-chip bandwidths (32 GBps), the reduce-scatter and all-gather phases dominate the total communication time. With HFP8, the time for all-gather phase is halved, as HFP8 broadcasts weights in 8 bits during this phase whereas FP8 shuffles reduced gradients in 16 bits. Also, as mentioned in Section 4, in HFP8, each learner updates weights for only $1/N^{th}$ of the total model which practically eliminates the time for the weight-update phase. It is noteworthy that our analysis includes the time it takes to compute the weight residual during the weight-update phase in HFP8. In effect, the communication time is decreased by 32%.

The benefits are enhanced when gradient compression is invoked. In Figure 9, we used a compression factor of $256\times$. This dramatically reduces the time for reduce-scatter, leaving the all-gather and weight-update as the key bottlenecks to performance. Since HFP8 addresses these components, the relative improvement in communication time is more pronounced (54%).

As the off-chip bandwidth is increased to 64 GBps and 128 GBps, the time consumed by the reduce-scatter and all-gather phases decreases, whereas the weight-update phase remains constant as it does not involve off-chip communication. Therefore, as a proportion, the weight-update phases occupies an increased fraction of the total time. Since HFP8 significantly alleviates this phase, the overall benefits are enhanced at higher bandwidths to 36% and 38% without gradient compression and 58% and 64% with gradient compression at 64 GBps and 128 GBps respectively.

# 5 Appendix E: Model Details and Convergence Curves for HFP8 Training

Figure 10: Convergence curves for AlexNet, ResNet50 and DenseNet between baseline and proposed FP8 schemes

## 5.1 MobileNetV2

We adapted a PyTorch implementation of MobileNetV2 and the standard pre-processing [15] of ImageNet ILSVRC2012 dataset. We used learning rate 0.0045 with a decay factor of 0.98 per epoch. For optimizer we used RMSPROP with $\varepsilon = 1$ as the original paper stated[13], Momentum of 0.9 and weight decay of 4e-5. The BatchNorm running mean and variance momentum is set to 0.1. For our purpose of studying quantization impact, we did not use weight averaging of 0.999 adopted by the original paper. The minibatch size is 256 over 8 V100 GPUs. We found that less than this $mb/GPU$ ratio would degrade the performance using Vanilla DataParallel module in PyTorch for distributed learning. Eventually we obtained the accuracy of 71.8% at epoch 270 with input size of 224 and width-multiplier of 1, agreeing with $71.8 - 72.0\%$ reported in the paper [14] and the Github repository [16]. The downsized MobileNetV2 with multiplier of 0.5 and 0.35 used the exact same parameters to compare with the full size MobileNetV2. The convergence is slightly worse than the reported results [16]. However since there is no other reports elsewhere of the 0.5 and 0.35 MobileNetV2 and we did not want to contaminate the experiment with different hyper parameters than 1.0 MobileNetV2, we used our own baseline for small MobileNetV2.

## 5.2 ResNets and DenseNet

We adapted ResNets v1 (including ResNet18 and ResNet50) and DenseNet(DenseNet121) from the models of torchvision [17] and the standard pre-processing of ImageNet ILSVRC2012 dataset. We used learning rate of 0.1 with a 0.1 decay of epoch 30,60,90 and 120, if applicable. For optimizer we used non-Nesterov SGD with momentum of 0.9 and weight decay of 1e-4. The BatchNorm

running mean and variance momentum is set to 0.1. The minibatch size is 256 over 8 V100GPUs. We found that 16mb/GPU if using 16GPU will degrade the final accuracy up to $1\%$, agreeing with the observation by others [18]. Eventually for ResNet50 we obtained 75.86 at Epoch 90 and 76.44 at Epoch 120. We used the baseline Ep120 FP32 model for Trans-Precision inference and the backbone model of MaskRCNN. Please see Fig.10 for results.

### 5.3 AlexNet

We adapted AlexNet model from torchvision [17] and the standard pre-processing of ImageNet ILSVRC2012 dataset. We used the learning rate of 0.01 with a 0.2 decay at epoch 20 and 30. For optimizer we used non-Nesterov SGD with momentum of 0.9 and weight decay of 1e-4. The minibatch size is 256 over 8 V100GPUs. At Epoch 45 we obtained the accuracy of $57.28\%$ matching the standard [19]. Please see Fig.10 for results.

### 5.4 LSTM PTB

We adapted the two-layer LSTM model with medium size (hidden dim of 650) from PyTorch Examples [21] on PennTreeBank dataset with the vocabulary size of 10,000. We adopted static learning rate 1.0 with 0.8 decay factor after Epoch 6. The minibatch size is 20 and the sequence length is 35. Eventually we obtained Validation and Test perplexity of 87.35 and 83.66, respectively.

### 5.5 Transformer Model

Figure 11: Convergence curves for Transformer base model on WMT 14 En-De

We adapted the implementation of FairSeq [22] and the setup of the Transformer Base model in the repository on the WMT 14 En-De Translation task. The choice of choosing the base model instead of the large model is for controlling training time, which is slowed down by $\times 7$ with our op-level quantization implemented. We used the Adam optimizer and modified the APEX package in FairSeq for HFP8. Detailed parameters can be found in the repository of FairSeq [23]. To calculate BLEU score vs. Epoch we used the script in the repository on the checkpoint generated at each epoch with beam 4, length penalty of 0.6, and removebpe option after compound splitting [24]. Eventually we obtained BLEU score 27.5 at Epoch 32 with 120,000 updates, in line with the publication [25]. For Trans-precision inference we used an averaged model over the last 5 epochs with BLEU score of 27.53.

### 5.6 Speech Model

We adapted the acoustic LSTM model of IBM speech [26] that contains 4 bi-directional LSTM layers and 2 fully-connected layers in the main network. Each LSTM layer contains 1024 cells with 512 on each direction. On top of the LSTM layers, there is a linear bottleneck layer with 256 hidden units followed by a SoftMax output layer with 32K units corresponding to CD HMM states. The

Figure 12: Convergence curves for 4-bidirectional-LSTM Speech in SWB300

LSTMs are unrolled 21 frames and trained with non-overlapping feature sub-sequences of that length. The 300-hour switchboard data set is used to train network. The training set consists of 262 hours of Switchboard 1 audio with transcripts. The test set is the 2.1-hour switchboard (SWB) data from 40 speakers. The batch size is 256. 4-gram language models (LMs) are used in decoding and acoustic weight is chosen as 0.05. The full precision baseline we obtained closely matches performance reported in [26].

## 5.7 MaskRCNN Models

We adapted a PyTorch implementation of MaskRCNN from [27], which follows the implementation structure of Detectron [28]. For reduced precision experiments, we used ResNet50 C4 backbone network pre-trained in PyTorch in FP32 precision. The MaskRCNN was trained and evaluated on COCO 2014 dataset [29]. The training was performed on 16 GPUs with effective batch size of 16 for 90k iterations. We used the standard SGD Optimizer with an initial learning rate of 0.02 which was decreased by 10 at 60k and 80k iteration. The details of the hyperparameter setting and data preprocessing can be found in the repository [27]. For the full precision baseline, we obtained a box (FastRCNN) AP (Average Precision that averaged over IoU thresholds) of 33.58 and a mask (MaskRCNN) AP of 29.27, which closely matches with the performance reported in [30].

Figure 13: Convergence curves for MaskRCNN on COCO

## 5.8 SSD-Lite Models

Figure 14: Convergence curves for MobileNetV2 SSD-Lite

We adapted a PyTorch implementation of SSD-Lite MobileNetV2 from [31]. The network was trained on a mixture of VOC2007 and VOC2012 *trainval* datasets and evaluated in VOC2007 testset [32]. We used MobileNetV2 [14] as the backbone network pre-trained using the ImageNet dataset. We replaced all the regular convolutions in SSD prediction layers with the inverted-residual blocks. The network has been trained for 200 epochs with a batch size of 32 and the cosine annealing ($t_{max} = 200$) with the initial learning rate of $0.01$. For optimizer, we used SGD with momentum $0.9$ and weight decay factor $0.0005$. Using one NVIDIA V100 gpu for training, we achieved $mAP$ of $68.8$, which matches with the performance reported in [31].

## 6 References