[Reviews · NeurIPS 2019]

Reviewer 1



Post rebuttal: I feel this is a strong paper and will maintain my score. - Figure 3 is a very nice visualisation; I hadn't thought of plotting the corruption to the objective landscape under model quantisation before. - This seems to out-perform some of the recent mixed-precision results as well; you may want to directly state this in one of the comparison tables - I'd like to see time comparisons for training and inference - there should be better baseline comparisons (although: this method seems to match normal training, so there's very little margin for it to be out-performed. The comparisons should be used to emphasize that more complex methods actually end up under-performing the proposed method)

Reviewer 2



After reviewing the feedback of authors and other reviewers, I've decided to keep my high score. This is a strong paper. -------------------------------------------- 1. Originality: 1.a. The idea of using 1-4-3 FP8 format is the evolution of prior work on using 1-5-2 FP8 format for training neural networks. 1.b. The idea and analysis of re-tuning batch normalization statistics to match 1-4-3 data is novel. 1.c. The idea of doing distributed training with FP16 reduce-scatter and HFP8 allgather is novel and is supported by an excellent analysis. 2. Quality: the paper provides a detailed analysis of hybrid FP8 training approach and highlights the benefits of having FP8 multiplication units that support both 1-4-3 and 1-5-2 formats. 3. Clarity: the paper is clearly written, and together with an appendix, provides enough information to reproduce the results. 4. Significance: the paper explores a practical technique to speed up training and inference of neural networks, at a cost of minimal model quality loss and introduction of new hardware primitives. While this paper builds upon previous work on FP8 training & inference, it provides significant improvements useful for the research community and the industry.

Reviewer 3



Originality - This basically amounts to using two different floating point formats - one for forward, and one for backward. Or another way to think about it is that we are allowing more freedom in the mantissa/exponent divide for floating point. That's a good observation to have, theoretically, but how would a framework implement this, practically? For example, maybe I missed it, but I don't see how you convert between 1-4-3 and 1-5-2 formats when you prepare for back prop if we were to productize this. Do the frameworks now have to support 2 more data types? Is the user aware of the data types, even? Quality - They did a sufficiently thorough study, including looking at impact on batch norm layers, thinking about weight update, etc. I appreciate the diversity of networks addressed as well - that's very convincing. Clarity - the paper is clearly written. How do you get away with FP16 master weights when most others need FP32? Significance - Is the intent to convince hardware vendors to provide this? Or is this for a custom chip? How does a reader take advantage of this? -------------------- After author feedback. The authors did not quite address my question about frameworks using this. I was referring more to annoying things like plumbing a new data type through every layer implementation. That is an often overlooked part that takes real work. However, despite that, I think this paper still passes the bar so I'll keep my score. My other issues were addressed.

[Author Response · NeurIPS 2019]

**Reviewer 1 Detailed comments**: Figure 3 is a very nice visualisation; I hadn't thought of plotting the corruption to the objective landscape under model quantisation before....This seems to out-perform some of the recent mixed-precision results as well; you may want to directly state this in one of the comparison tables

**Author response:** We thank the reviewer for the comments. After our submission, we noted a new 8-bit mixed precision work showing up [arxiv.org/abs/1905.12334] using FP152 accumulated in FP32 and we'll add that to our comparison.

**Detailed comments**: I'd like to see time comparisons for training and inference

+**Improvements**:comparison in training times to standard FP32 + inference times (although I appreciate that GPUs/TPUs don't support these representations so this would require implementation on FPGAs which would be a major ask)

**Author response**: The speed-up from FP32 to FP8 strongly depends on the chip architecture and any additional compiler and software optimizations. Comparisons are in general quite tricky—especially since architecture can be optimized around a different precision point. In our hardware (ASIC) experiments, we've seen a $\sim \times 2 - \times 2.5$ boost in peak performance moving from FP16 to FP8. While our hardware does not target for FP32, we refer to a $\times 8$ peak performance boost from FP32 to FP16 from the Nvidia Tensorcore, leading to an estimated $\times 16$ improvement from FP32 to FP8. Intel FPGA has shown $\times 10$ boost from FP32 to FP8 in peak throughput[Gordon Chiu et al. ISPD'18].

**Detailed comments**: there should be better baseline comparisons (although: this method seems to match normal training, so there's very little margin for it to be out-performed. The comparisons should be used to emphasize that more complex methods actually end up under-performing the proposed method)

+**Improvement**:baselines going beyond the 1-5-2 format

**Author response**: We agree with the reviewer that our method is intended to match normal training. Complex methods may require changes from FP32 models and training/inference scripts, e.g., introducing a quantization-friendly normalization. Such a wide design space is beyond the scope of this paper, since it could impose extra burdens on users to modify/calibrate their models, hyperparameters and optimizers. Our FP8 training/inference scheme requires minimal effort from the users as no changes to the network architecture, data pre-processing, or hyperparameters are needed.

**Reviewer 2 Improvements**: It would be helpful to clarify data formats of each step in Table 2.

It would be helpful to clarify that the weight update is applied to 1/N of weights in Table 3.

**Author response**: We thank the reviewer for the comments. Table 2 and 3 will be updated.

**Reviewer 3 Detailed comments**:... but how would a framework implement this, practically? For example, maybe I missed it, but I don't see how you convert between 1-4-3 and 1-5-2 formats when you prepare for back prop if we were to productize this. Do the frameworks now have to support 2 more data types? Is the user aware of the data types, even?

**Author response**:We thank the reviewer for the comments. From a hardware perspective, mixed 8-bit precision operations such as convolution of FP143 weights or activations with FP152 gradients is practical, easy to implement and only costs around 5% additional area in the floating point engines (FPU) as stated in line 175. This stems from the fact that our FPU design can take hybrid 8-bit inputs—FP143 and FP152 operands and produce products in FP169—requiring no conversion between FP143 and FP152 formats. More details on the FPU and hardware architecture will be discussed in future hardware conferences. From a framework perspective, we intend to map the forward, backward and update computations directly to the right Hybrid-FP8 libraries that we provide along with the hardware (in the graph optimizer of the framework)—if the user expresses an interest in enabling FP8 operations at the Python level. We intend to automatically choose these formats, but also give users the option to enable/disable these features.

**Detailed comments**:How do you get away with FP16 master weights when most others need FP32?

**Author response**: The FP32 master copy of weights was needed to avoid the swamping [2,23] problem due to insufficient mantissa bits in the Weight Update step. To overcome this, we keep a copy of the quantization error(residual) instead of the weight itself in FP169(Table 3). Since the residual is small, its exponent bit will adjust to store information in addition to the 9-bit mantissa. In total, we estimated at least 14 mantissa bits of the original weights are preserved after combining the FP143 weight and FP169 residual, sufficient to avoid swamping. With this trick, we're able to get away without using the FP32 master copy of weights.

**Detailed Comments**:Is the intent to convince hardware vendors to provide this? Or is this for a custom chip? How does a reader take advantage of this?

**Author Response**: Both are possibilities. We intend to promote 8-bit floating point solutions agnostic to specific hardware. Readers could use our learning to improve next-gen training and inference hardware platforms. Our theoretical learning of quantization also has universal value for the quantization research community.

**Improvements**: It seems like the precision for the layers is very carefully chosen empirically. How would a user use this in the general case, training a model from scratch, without having to add yet more hyperparameters?

**Author Response**:We agree with the reviewer that generally some level of empiricism exists in quantization, which has motivated us to cover a wide-range models and complex datasets. This study has revealed that a fixed set of FP8 rules could be applied and work universally well across a wide spectrum of models and datasets. Given how well these rules work, we don't anticipate requiring the user to specify any new hyperparameters. From a gradient perspective, we also adopted automatic loss-scaling techniques such as APEX to autoscale the dynamic range of gradients—eliminating the need to handpick the loss scaling factor.

[Meta-Review · NeurIPS 2019]

This paper demonstrates a method for low precision 8-bit deep neural network training without loss of performance across a number of popular tasks, including classification, speech recognition, and translation. The reviewers agree that this is a strong paper with solid analysis and a number of novel contributions to low precision training.